# Dermatologic Manifestations of Neurofibromatosis Type 1 and Emerging Treatments

**DOI:** 10.3390/cancers15102770

**Published:** 2023-05-16

**Authors:** Dina Poplausky, Jade N. Young, Hansen Tai, Ryan Rivera-Oyola, Nicholas Gulati, Rebecca M. Brown

**Affiliations:** 1Department of Dermatology, Icahn School of Medicine at Mount Sinai, New York, NY 10029, USA; 2Department of Neurology, Icahn School of Medicine at Mount Sinai, New York, NY 10029, USA

**Keywords:** neurofibromatosis type 1, neurofibroma, cutaneous, NF1, MPNST, cutaneous neurofibroma, plexiform neurofibroma, JCMML, glomus tumor

## Abstract

**Simple Summary:**

Neurofibromatosis type 1 (NF1) is a genetic neurophakomatosis (neuroectodermal disorder) associated with a wide array of skin findings. The most recognizable feature is cutaneous neurofibromas; however, less common dermatologic stigmata may also occur. Currently, there are limited therapeutic options for the cutaneous manifestations of NF1. This review summarizes the dermatologic sequelae of NF1 and provides an update on emerging treatments. Appropriate care of these patients often requires interdisciplinary collaboration between neurologists, oncologists, and dermatologists, among other specialties. Increased awareness of these conditions is crucial for providing adequate care to patients with NF1.

**Abstract:**

Neurofibromatosis type 1 (NF1) is an autosomal dominant tumor predisposition syndrome that increases one’s risk for both benign and malignant tumors. NF1 affects every organ in the body, but the most distinctive symptoms that are often the most bothersome to patients are the cutaneous manifestations, which can be unsightly, cause pain or pruritus, and have limited therapeutic options. In an effort to increase awareness of lesser-known dermatologic associations and to promote multidisciplinary care, we conducted a narrative review to shed light on dermatologic associations of NF1 as well as emerging treatment options. Topics covered include cutaneous neurofibromas, plexiform neurofibromas, diffuse neurofibromas, distinct nodular lesions, malignant peripheral nerve sheath tumors, glomus tumors, juvenile xanthogranulomas, skin cancer, and cutaneous T-cell lymphoma.

## 1. Introduction

Neurofibromatosis type 1 (NF1) is an autosomal dominant genetic tumor predisposition disorder that affects approximately 1 in 3000 people without gender or race predilection [1]. It is a clinical diagnosis that requires at least two of the following: a first-degree relative (parent or child) with NF1; six or more café-au-lait macules (>5 mm in pre-pubertal and >1.5 cm in post-pubertal patients); axillary and inguinal freckling; two or more hamartomatous Lisch nodules of the iris; optic pathway glioma; two or more neurofibromas of any type or one plexiform neurofibroma (PN); disease-typical bony dysplasia; or a known pathologic mutation in the NF1 gene. Patients with NF1 are at greater risk of cardiovascular, musculoskeletal, and nervous system abnormalities, in addition to being prone to benign and malignant tumor development throughout their life [2].

NF1 is caused by a zygotic loss-of-function mutation in one allele of the NF1 gene, resulting in a 50% reduction in functional neurofibromin in all cells of the body. Neurofibromin is a ubiquitously expressed tumor suppressor protein, but is present in the highest concentrations in the central and peripheral nervous system, especially in neurons, astrocytes, oligodendrocytes, and Schwann cells. It is a tumor suppressor protein that primarily functions to downregulate Ras [3]. Ras is a GDPase signal transduction protein that activates the downstream mitogen-activated protein kinase (MAPK) pathway (Ras/Raf/MEK/Erk signaling) and the PI3K/Akt/mammalian target of rapamycin (mTOR) cell proliferation pathway. Other identified mechanisms for tumor suppression by neurofibromin include the upregulation of adenylyl cyclase, pro-apoptotic effects via Ras signaling [4], regulation of cell adhesion and motility by interacting with focal adhesion kinase (FAK) [2], suppression of epithelial mesenchymal transition (EMT) by downregulating EMT-related transcription factors [5], and suppression of heat shock factor 1 [2]. Lacking adequate functional neurofibromin in the Schwann cell progenitor pathway, NF1 patients are at higher risk for tumors including malignant peripheral nerve sheath tumors (MPNSTs), optic pathway gliomas (OPGs), rhabdomyosarcomas, neuroblastomas, gastrointestinal stromal tumors (GISTs), pheochromocytomas, carcinoid tumors, and breast cancer, among others. Somatic NF1 mutations also occur in sporadic cancers including brain, lung, breast, ovarian, and melanoma [2].

About half of all NF1 cases are inherited, and half result from de novo, spontaneous NF1 mutations. Genetic testing is not required for a diagnosis of NF1, but any detected heterozygous pathologic variant in the NF1 gene can serve as one of the two required minimum criteria for a diagnosis of NF1 [6]. The condition has 100% penetrance, but phenotypic expression is highly variable, even within one family [7]. The first NF1 loss-of-function event is inherited or acquired as a germline mutation, and the second is a somatic event typically occurring within the neural crest-derivative population [2]. Here, we present a narrative review of the dermatologic manifestations and associations of NF1, along with their respective emerging treatment paradigms.

## 2. Cutaneous Neurofibroma

Cutaneous neurofibromas (CNs) are physically deforming tumors with self-limiting growth arising in the skin of NF1 patients (Figure 1). The tumors begin appearing in late childhood or early adolescence. Histopathologically, they are composed of a complex conglomerate of multiple cell types, including but not limited to neoplastic cells lacking both NF1 alleles and haploinsufficient endoneurial cells, fibroblasts, macrophages, T lymphocytes, and mast cells [8]. Only the Schwann-derived neoplastic cells lack both NF1 alleles, but support cells play a key role in the genesis and growth of these benign tumors. Tumors start appearing in the peripubertal period in 85–95% of NF1 patients and never regress. This leads to a high degree of psychological distress and socioeconomic bias associated with the degree of disease burden, and contributes to a loss of income and agoraphobic behaviors in adults. People with NF1 often cite their skin tumors as a primary negative contributor to their mental health. Investigation into the psychological ramifications of NF1 in children and adolescents has revealed compromised self-image, discordance between the quality of life projected by caretakers and the patient, and greater anxiety than healthy controls [9]. Similarly, in adults, significant correlations have been identified between NF1-affected patients and decreased quality of life, body image, sexual self-esteem, and the presence of depressive symptoms [10]. CNs can cause pruritus (sometimes associated with cutaneous mastocytosis), stinging, burning, tenderness, and even bleeding [11,12].

CNs can appear and grow over time; however, little is known about how hormonal shifts impact changes in CN tumorigenesis or growth. Commonly cited evidence supporting the role of hormones in CN development states that they begin to appear in late childhood/early adolescence, around the time of puberty. These are correlative data at best, and do not explain why there are no sex differences in the severity of CN burden between males and females. Most studies investigating the role of hormones focus on pregnancy and are largely based on anecdotal evidence or subjective recall. In a survey study evaluating 105 primigravida NF1 patients, approximately half retrospectively reported enlargement of existing neurofibromas and increased growth of new neurofibromas [13]. Recall bias is a likely confounder of these data. While it has been hypothesized that CN tumorigenesis and growth during pregnancy is attributable to hormonal changes in progesterone, as CNs have been shown to express progesterone receptors [14], the association is likely scientifically reductionist given that there are many physiologic changes in pregnancy including hormones, immunomodulation, and edema. Additionally, no prospective studies have been performed, and cross-sectional studies are statistically limited by the highly variable inter-individual degree of cutaneous involvement of NF1. Significant neurofibroma growth has been reported in a single individual receiving high-dose depot progesterone [15], but a high percentage of women and girls with NF1 are using oral contraceptives without measurable recorded changes in cutaneous neurofibroma growth rates. A retrospective review evaluated 13 NF1 patients before and after pregnancy and compared them to age-matched nulliparous, nulligravida females with NF1. CNs were measured with calipers and quantified using the sum of longest diameters (SLD). There were no significant differences in overall tumor volume or significant increases in SLD in pregnant patients versus the control group. However, significantly more pregnant patients subjectively reported growth of CNs compared to controls [16]. Further research is necessary, ideally utilizing novel techniques to quantify CN tumor burden in individuals before, during, and after pregnancy. Murine models utilizing oophorectomized females and orchiectomized males may also reveal differences between the sexes with balanced hormone supplementation in genetically engineered models.

The management of CNs is a challenge for patients and providers, alike. The tumors reside below the tight junctions of the stratum corneum, separating the epidermis from the dermis, resulting in impeded tissue penetration of topical agents. The only gold-standard treatment is physical removal of the tumor, and even then, there is a risk of recurrence if inadequately resected, and also at the site of suture penetration of the skin (potentially due to activation of injury-responsive pluripotent Schwann cell precursors). Scarring occurs based on technique, which sometimes obviates the patient’s desire for decreased physical stigmata of CNs. Although many topical and systemic therapies and procedures with varying levels of efficacy have been explored, there are currently no Food and Drug Administration (FDA)-approved pharmacotherapies for CNs. Identifying an effective pharmacotherapy is constrained by a need for long-term safety and tolerability in patients who will experience ongoing accumulation of lesions throughout their lifetime. Despite this fact, the treatment of CNs should remain a top priority in the exploration of new advances in NF1.

Although there is currently no standard procedure or medication for CN removal, several avenues have been explored within the literature, including topical and oral medications, surgical removal, and laser and light-based therapies. To counteract Ras hyperactivity, the downstream mTOR pathway has been targeted pharmacologically to investigate any efficacy in reducing CN size. mTOR activation leads to overall increased protein synthesis and cell proliferation, and likely plays a role in CN tumor formation [17].

## 3. Systemic Therapies

The earliest tested immunotherapy for NF1-associated tumors was imatinib mesylate, a C-Kit receptor tyrosine kinase inhibitor targeting mast-tumor cell signaling, for the treatment of plexiform neurofibromas. It was subsequently also tested against CNs, at doses of 400 mg orally twice daily or once daily. Small reductions in the visual burden of CNs were noted at the expense of dose-limiting systemic side effects including fluid retention, nausea, vomiting, myalgias, fatigue, skin rash, diarrhea, and hepatotoxicity [18].

In a 2014 trial, researchers studied the efficacy of the mTOR inhibitor sirolimus (rapamycin) in the management of NF1-associated PNs [19]. Everolimus, a rapamycin derivative that inhibits mTOR, had been previously shown to inhibit the proliferation of neurofibromin-deficient cells in an in vitro cell culture model [17]. Slopis et al. went on to examine the efficacy of everolimus for CNs in a 2018 clinical trial, completing one of the largest trials for CNs to date. The study included adult patients with NF1 and CNs. Everolimus was started at 10 mg daily and up- or down-titrated based on serum concentrations to maintain 5–15 ng/mL for 6 months. Twenty-two participants were treated in the study, with five patients withdrawing due to adverse events, most commonly stomatitis, upper respiratory infections, skin irritation, and gastrointestinal upset, leaving a total of seventeen patients who completed the trial. The researchers found a statistically significant decrease in the absolute value of paired lesion height (*p* = 0.048) and a reduction in the percent change in average height for paired lesions approaching significance (*p* = 0.080). Three out of 17 participants were found to have significant reduction in the lesion surface volume >2 standard errors below the baseline volume. For paired lesions, the absolute change in surface volume and the percent change in surface volume were not statistically significant (*p* = 0.582 and *p* = 0.425, respectively). Overall, the study was limited by sample size and withdrawal due to adverse events [17]. One conclusion is that reduction in the size of CN is possible, but that a more tolerable approach is necessary.

Selumetinib, a MEK1/2 inhibitor that acts on the MAPK pathway, was shown to reduce the size of PN by at least 20% in 71% of pediatric participants. In 2020, selumetinib (Koselugo; AstraZeneca) became the first and only FDA-approved medication for the indication of unresectable PN in the pediatric NF1 population SPRINT study [20,21,22]. However, the ability for selumetinib’s clinical effect to cross over from PN to CN treatment has not yet been established. There is an ongoing clinical trial to examine the effect of oral selumetinib on CN burden in adults (NCT02839720) [23].

## 4. Topical Treatments

Given the heavy burden of adverse events attributable to systemic medications targeting tyrosine kinase inhibitors or Ras/mTOR pathway signaling molecules, a topical option would be ideal for NF1 patients. In 2009, a clinical trial (NCT00865644) investigated imiquimod, an immunomodulator that acts as a Toll-like receptor 7 (TLR-7) agonist, for the treatment of CNs. However, it failed to produce a significant reduction in tumor volume, and minimal skin inflammation was seen with prolonged treatment, suggesting poor activation of the immune response [24,25]. A double-blind randomized controlled trial investigating the gel NFX-179 (NFlection Therapeutics, Inc., Boston, MA, USA), a topical MEK inhibitor, against the gel vehicle was completed in 2021. The complete statistical analysis is still pending. The concentrations 0.05%, 0.15%, and 0.50% demonstrated −1.6, −11.9, and −16.7, percent changes in CN volume, respectively, after 28 days of once daily application [26]. An additional clinical trial testing NFX-179 at higher concentrations, 0.50% and 1.5%, against the gel vehicle is ongoing (NCT05005845) [27]. An open-label single-arm phase 1 trial is examining the effect of topical diphencyprone, a hapten-based ointment that instigates a type IV hypersensitivity reaction and targets T lymphocytes to attack the CN tumor environment (NCT05438290) [28]. Table 1 summarizes the systemic and topical treatments for CNs.

## 5. Surgical Intervention

Many techniques have been explored for the removal of CNs. Chamseddin et al. developed a surgical approach that emphasizes the adequate evacuation of deep dermal tumors, where a significant portion of the CNs reside [29]. The technique utilizes local anesthesia, shaving the outer pedunculated portion of the tumor with a dermablade or razor blade, then evacuating the remaining collagenous dermal portion of the tumor and excising it from normal skin. The wound is then closed via primary intention with interrupted sutures. This technique was developed, in part, out of the recognition that a widely employable surgical technique requiring minimal supplies was necessary to ensure adequate access to care for the global NF1 community. In the study, 12 patients participated, with a total of 83 tumors removed. Patients had up to ten CNs removed per session. Patients’ quality of life index was measured before and after, with significant improvement in CN symptoms, ability to participate in daily activities, and personal relationships (*p* < 0.001). The researchers noted that patients were, overall, very satisfied with the aesthetic outcome and the treatment experience. This is in contrast to more traditional surgical approaches that require general anesthesia, highly trained surgical specialists, and sterile surgical fields, and have a high-risk side effect profile [29]. Indeed, given the demand for multiple resections throughout a patient’s lifetime, physicians should maintain a degree of concern over the cumulative risk of repeat general anesthesia for NF1 patients.

## 6. Electrodessication

Electrodessication is an alternative ablative technique that can be performed either with local or general anesthesia depending on the extent of treatment. Lutterodt et al. evaluated patient satisfaction with electrodessication of CNs. Six patients underwent electrodessication of up to hundreds of CNs per session under general anesthesia. Overall, adverse events were minimal, with one minor wound infection and one case of minor bleeding. Five of the six patients stated they preferred electrodessication to surgical excision [30]. In the authors’ experience, electrodessication treatment of large numbers of tumors can lead to intolerable post-procedural pain in some patients, but this has not been reported in the literature.

## 7. Laser-Based Interventions

Although surgical CN removal typically yields minimal scarring and high patient satisfaction, resection may not be practical or desired by all patients due to the risk of regrowth and induction of a new CN. Given the procedure time, there is a limit to how many CNs can be removed in one clinic session [29]. Alternatively, several studies have examined the efficacy of laser and light-based therapies to decrease the size and appearance of small CNs ~1 cm or less. Since 1987, carbon dioxide (CO_2_) laser therapy has been used to treat CNs with comparable results to excision [31,32,33]. Similar to surgical removal, the removal of large numbers of CNs can be accomplished with a CO_2_ laser under general anesthesia. Overall, patients undergoing this widespread removal reported being pleased with the results, with 8 of 11 patients stating that they desired additional rounds of treatment. Compared to surgical excision, 7 of 11 patients stated that the scarring was superior in their subjective approximation. Potential downsides include post-operative pain (2 of 11 patients) and pruritus (9 of 11 patients) [34,35].

Another device used in the treatment of CNs is an erbium:yttrium-aluminum-garnet (er:YAG) laser, which is an ablative laser similar to CO_2_, but which utilizes shorter wavelengths that permeate tumor cells more facilely [36]. The laser emits light energy that is absorbed by water molecules, resulting in tissue vaporization and thermal injury, and subsequent collagen production [37]. This laser is associated with reduced collateral thermal damage compared to CO_2_ lasers, thus maximizing wound healing and aesthetic outcome. One study examined the histologic differences in tissue necrosis and tumor removal between er:YAG and CO_2_ lasers for 126 CN specimens. Lesions were vaporized with either CO_2_ or er:YAG laser in vivo and then excised for histological examination. Both modalities of laser completely removed tumor tissue; however, upon histopathological analysis, the er:YAG laser produced less surrounding tissue damage and decreased post-operative healing time. The authors hypothesize that reduced damage by the er:YAG laser is a result of enhanced photothermal selectivity. They also compared the cosmetic outcomes and scar formations of CO_2_ and er:YAG laser ablation for the treatment of CNs. Participant Fitzpatrick skin phototypes ranged from II to IV. Scars were assessed four months postoperatively by a blinded panel of five healthcare professionals unrelated to the study. The appearance of scars was graded on a scale of 1–6 (1 = excellent to 6 = unacceptable) for the following categories: color, contour, matte, distortion, and overall scar assessment. Three hundred and ten scars in 12 patients were assessed. While additional studies are needed, and the cost and availability of the technique must be taken into consideration, these data provide important information for clinical counseling in discussing treatment options, where available.

One study reported the use of a neodymium-doped:yttrium-aluminum-garnet (Nd:YAG) laser to treat CNs. This laser utilizes a similar methodology to the other aforementioned lasers, in that it delivers heat energy. The water molecules of a CN absorb the output from the Nd:YAG laser, generating a supraphysiologic thermal response and causing tumor denaturation. Kim et al. report on a case of satisfactory CN removal outcome using shave removal followed by 1444 nm Nd:YAG photocoagulation. Following removal, the lesion was excised and stained for S100, which demonstrated reduced neural cells in the post-procedural sample. The patient reported satisfaction with the cosmetic and functional outcomes and rated laser-assisted removal as more convenient than traditional excision [38]. There is an ongoing clinical trial evaluating the tolerability and effectiveness of 980 nm and 755 nm lasers (NCT04730583) [39].

## 8. Light-Based Interventions

Tissue-sparing light-based techniques have also been utilized in the treatment of CNs, including photodynamic therapy (PDT) with the inert chemical 5-aminolevulinic acid (5-ALA). 5-ALA is absorbed by cells and is converted to protoporphyrin IX (PpIX). When PpIX is exposed to light, it generates reactive oxygen species, which induce apoptosis. Quirk et al. conducted a study wherein CNs were treated with microneedle-assisted delivery of 5-ALA (*n* = 14) and vehicle (*n* = 4). Eighteen hours later, CNs were irradiated with 633 nm frequency red light via PDT at up to a maximum tolerable dose of 100 mJ/cm^2^. Forty-eight hours later, tumor biopsies were performed. 5-ALA-treated tumors demonstrated mixed inflammatory infiltrate and a higher concentration of apoptotic cells (*p* = 0.002) than vehicle-treated tumors. Mature CNs consist of inactive fibrous tissue, and this technique requires metabolic uptake of 5-ALA. Therefore, the authors of this study hypothesize that this technique may be best suited to treat new metabolically active tumors rather than mature, fibrotic CNs. Although this approach has promise as a normal tissue-sparing technique, the authors did not investigate patient-reported outcomes or see a significant reduction in CN size [40]. A summary of procedural treatment options for CNs is outlined in Table 2.

## 9. Plexiform, Diffuse, and Distinct Nodular Neurofibromas

PNs frequently involve the skin, although an exact percentage is unknown. PNs can infiltrate into all three tissue layers, and when they affect the dermis, the appearance is classically rugated, velvety soft to the touch, and of dusky or erythematous color. Approximately 50% of patients with NF1 develop PNs [43]. In contrast to CNs, which are benign and have no malignant potential, there is an 8–13% lifetime risk of malignant transformation of PNs to an MPNST in NF1 patients. PNs are benign tumors derived from the Schwann cell precursor population that develop within large nerves and nerve plexi, as opposed to the dermal or subdermal location of CNs [44]. PNs arise in deep tissues; however, they can invade into the dermis, and, in such cases, should be distinguished from CNs and monitored closely with magnetic resonance imaging (MRI) to detect potential malignant transformation [45,46]. The cell of origin of PNs is likely embryonic, with loss of the second NF1 allele occurring early in the neural crest cell ontogeny, resulting in retention of multipotency throughout the patient’s lifespan. The tumors have an unpredictable growth rate, with periods of relative growth and possibly regression, although it is likely that an accurate measure of growth rate has been stymied by the technological limitations associated with volumetric delineation of three-dimensionally complex tumors that are frequently colliding with neighboring PNs. Loss of the Cyclin-Dependent Kinase 2a and 2b (CDKN2A2B) is a common transitional event permitting conversion from a PN to an atypical neurofibromatous neoplasm of uncertain biologic potential (ANNUBP), an intermediately aggressive tumor lying on the spectrum between benign PN and malignant MPNST. Loss of the Polycomb Repressor Complex 2 (PRC2) genes are most instrumental in enacting malignant transformation to MPNST [44]. MEK inhibitors are currently being investigated for PNs. Selumetinib is now FDA-approved for unresectable symptomatic PN in children with NF1, and selumetinib or other MEK inhibitors are occasionally offered off-label to adults. Clinical trials investigating other MEK inhibitors, namely mirdametinib in adolescents and trametinib in pediatric populations, for PNs found that 42% and at least 50% of patients, respectively, exhibited a >20% decrease in target PN tumor volume [43]. There is currently one ongoing clinical trial to determine the efficacy and safety of binimetinib for the treatment of PNs in children and adults (NCT03231306) [35].

Diffuse neurofibromas present as plaque-like induration of the skin, most frequently in the head and neck, noted in young adult patients. The potential for sarcomatous conversion of these tumors is low; however, cases have been reported in the literature [47]. Another subtype of neurofibroma is the distinct nodular lesion (DNL). These benign tumors have a simple fusiform morphology and appear only within the sheath of a single peripheral nerve. They may arise later in life, in adulthood, as compared with the congenital origins of true plexiform tumors. Resection is typically more facile than with plexiform tumors due to the relative isolation of the tumor, but as with all nerve sheath tumors, the nerve must be sacrificed for complete resection [48]. In methylation profiling, CNs, DNLs, and PNs (and NF1-associated ganglioneuromas) all showed distinct methylation profiles. ANNUBPs and low-grade MPNSTs had indistinguishable methylation profiles [49].

## 10. Malignant Peripheral Nerve Sheath Tumor

MPNSTs are rare, aggressive, and invasive cancers that comprise approximately 10% of all soft tissue sarcomas, and are primarily associated with NF1 or with radiation exposure in sporadic cases [50]. In clinical practice, they are rarely associated with dermal features of NF1, although as with PNs, the exact percentage has not yet been documented. MPNST involving the skin (Figure 2) may be detected and treated earlier than deep tissue MPNST due to difficult-to-miss malformation and ulceration. They form a molecularly diverse type of tumor, ranging between pauci-mutational genetics and widespread chromothripsis, or elevated tumor mutation burden [51,52]. MPNSTs confer a poor prognostic outcome and represent the leading cause of death in patients with NF1 [50]. Approximately 60% of MPNSTs lose the congenital NF1 mutation during malignant conversion, although in some percentage of cases in the earlier literature, the NF1 mutation may have been undetectable due to technological limitations and the fact that the NF1 gene is very long and complex [53]. Evans et al. studied NF1 patients in two population registries from 1984 to 1996. They found that NF1 patients developed MPNST at an annual incidence of 1.6 per 1000 and have an 8–13% cumulative lifetime risk of developing an MPNST. When compared to patients with sporadic MPNST, those with NF1 were diagnosed at a statistically significantly younger age (median 26), compared to age 62 (*p* < 0.001). Despite being more likely to be diagnosed at a younger age, the five-year survival rate is lower (21%) than for those with sporadic tumors (42%) (*p* = 0.09) [54]. This is potentially related to delayed diagnosis in a patient population living with multiple painful tumors who are less likely to recognize the signs of malignancy. Treatment for MPNST is primarily surgery, with the best outcomes for tumors located in the limbs rather than the trunk. Complete resection is frequently not feasible due to tumor bulk and proximity to essential nerves. Biopsies result in a high false-negative rate attributable to the topographic complexity of a large tumor within which malignant conversion may only occur in a small region. Systemic chemotherapy and ionizing radiation have a limited role and are reserved for high-risk tumors. There are no FDA-approved treatments available. Multiple clinical trials for MPNST treatment are currently underway, including one investigating sirolimus and selumetinib (NCT03433183), the multi-kinase inhibitor PLX3397 in combination with rapamycin (NCT02584647), and ribociclib, a CDK4/6 inhibitor (NCT03009201) [50].

## 11. Glomus Tumor

In addition to its innocuous hallmark skin findings, including café-au-lait macules, axillary and inguinal freckling, and benign CNs [55], glomus tumors represent a less-recognized association of NF1. Glomus tumors are rare, typically benign neoplasms arising from small arteriovenous anastamoses and modified smooth muscle cells within the thermoregulatory glomus bodies. Glomus bodies are concentrated within the dermis of the fingers and toes. They are predominantly present in palmar and plantar skin, particularly in the subungual region of the fingertips (Figure 3) [56,57]. Although these tumors are typically benign, there is a ~3% risk of malignancy, and an additional 3.6% have uncertain malignant potential [58]. The risk of malignant glomus tumors in NF1 patients is currently unknown. The cellular origin of these tumors is poorly understood [59]. Due to their rarity, glomus tumors are under-recognized, delaying the diagnosis to an average of approximately 10 years from symptom onset to diagnosis [57]. The typical triad of signs is localized tenderness, severe paroxysmal pain, and increased cold sensitivity. In addition to being under-recognized, glomus tumors confer significant morbidity associated with pain and dysfunction [57]. On physical examination, the nail and pulp of the affected fingertip may appear normal, but may alternatively manifest as blue nail discoloration or nail deformation. The diagnosis is clinical, although high-resolution MRI or plain radiographs can reveal bony defects [57].

Multiple case reports of glomus tumors in patients with NF1 have been reported in the literature [61]. It is estimated that up to 5% of adult NF1 patients have a glomus tumor [57]. Kumar et al. conducted a retrospective cohort study and evaluated 42 glomus tumors in 34 patients. Twelve tumors were found in six patients with NF1, while 30 tumors were found in 28 control participants. Recurrence and multifocal tumors were more common in NF1 patients; however, the difference was not statistically significant [62]. Harrison et al. conducted a retrospective chart review evaluating 21 patients diagnosed with a glomus tumor between 1995 and 2010. Six of the 21 subjects had NF1. There was a statistically significant association between glomus tumor and neurofibromatosis diagnosis, with an odds ratio of 168:1 (*p* < 0.001) [63]. Another systematic review studied 36 patients with neurofibromatosis and glomus tumors from 1938 to 2013. The sex predilection, tumor location, and tumor burden were found to be similar to non-NF1 patients with sporadic glomus tumors. In both populations, these tumors are more common in women, typically in a subungual location, and solitary [64]. NF1 patients are more likely to have multifocal and recurrent tumors [56].

The pathoetiology of this association is not well defined. One study evaluated 20 glomus tumors among 11 NF1 patients and compared them to two sporadic, non-NF1-associated, glomus tumors. Somatic and germline mutation analysis of NF1-associated glomus tumors showed inactivation of both NF1 alleles within the alpha-smooth muscle, actin-positive glomus cells. In these cells, there was increased activation of the Ras-MAPK pathway, suggesting that the pathogenesis of glomus tumors in NF1 may be due, in part, to the post-transcriptional loss of neurofibromin function. No abnormalities in the NF1 gene or Ras-MAPK activation were found in sporadic glomus tumors [65]. NF1 is also associated with significant vasculopathies, particularly renal artery stenosis, which can be attributed to a similar mechanism. The loss of neurofibromin in vascular smooth muscle cells leads to the neointimal proliferation of smooth muscle and subsequent vasculopathy [66].

Treatment includes surgical excision, via a direct trans-ungual or lateral subperiosteal approach. These procedures are performed under local anesthesia and typically without complications [57]. Due to the rarity of a malignant glomus tumor, there are limited data for therapeutic strategies. First-line treatment includes wide resection with a goal of negative margins. Metastasis is rare and indicates a poor prognosis. Adjuvant therapy options in these cases include radiation therapy and chemotherapy [67]. Tumor recurrence is possible; however, the rate is variable within the literature from 0 to 33.3%. Additionally, patients can develop complex regional pain syndrome (CRPS) even after surgical excision [57]. Given that neurofibromin plays a role in nociceptive sensory neuron regulation, it is hypothesized that NF1 patients may be at increased risk of developing pain from these tumors [61]. Pain prevalence in NF1 patients is currently being investigated. Sanagoo et al. conducted a systematic review and meta-analysis on the quality of life in NF1 patients. They found that NF1 patients had significantly higher bodily pain scores compared to controls without NF1 [68]. Glomus tumors are often symptomatic, have delayed diagnoses, and can be malignant, and, therefore, NF1 patients should be regularly screened for these tumors by their dermatologist.

## 12. Juvenile Xanthogranuloma

Another lesser-known cutaneous association with NF1 is juvenile xanthogranuloma (JXG). JXG is the most common type of non-Langerhans cell histiocytosis. It presents as an asymptomatic yellow papule, typically on the head and neck. They are typically solitary lesions; however, patients can present with multiple [69]. They appear in childhood, often within the first year of life [70], and can be seen in 15–35% of children with NF1 [55]. They are benign and self-limiting, and often spontaneously regress. In the general population, JXG incidence has been associated with a risk of juvenile myelomonocytic leukemia (JMML) [70]. Independent of JXGs, NF1 patients have an increased risk of developing JMML. Approximately 4–10% of patients with JMML also have a diagnosis of NF1. The loss of neurofibromin leading to Ras hyperactivity is critical to the pathogenesis of JMML. There are conflicting data in the literature about the increased risk of JMML in NF1 patients with JXG. One retrospective case–control study found that JXG does not confer a further increased risk of malignancy in patients with NF1 [71]. Given that they often spontaneously resolve, there is no treatment for JXG. Surgical excisions may be performed for cosmetic purposes [70].

## 
13. Skin
Cancer


Less prototypical but equally important cutaneous findings include melanoma and non-melanomatous skin cancers (NMSC) [55]. As a tumor suppressor protein, NF1 is also somatically inactivated in multiple malignancies, including melanoma. NF1-mutated cutaneous melanoma comprises about 10–15% of all melanoma cases, while about 25% are Ras-mutated [72]. NF1-inactivating mutations are found in acral, mucosal, desmoplastic, and UV-associated skin melanomas [73,74,75].

In a study analyzing 4122 patients with NF1 compared to 41,064 patients without NF1, NF1 patients had significantly increased odds of basal cell carcinoma (BCC) (Odds ratio (OR): 1.30; 95% CI: 1.10–1.53; *p* = 0.002), SCC (OR: 1.32; 95% CI: 1.07–1.63; *p* = 0.008), and melanoma (OR: 2.27; 95% CI: 1.75–2.93; *p* < 0.001) [76]. A separate study compared rates of a range of tumors in 6739 NF1 patients with a comparison cohort. The rate ratio (RR) of malignant melanoma was 3.6 (CI: 2.2–5.6; *p* < 0.001) in NF1 patients, and the NMSC RR was 1.6 (CI: 1.2–2.0; *p* < 0.002) [77]. In addition, both Miraglia et al. and Landry et al. found melanoma tumors in those with NF1 to have a higher Breslow depth (3.2 and 2.7 mm, respectively) than sporadic melanomas (1.5 mm) [78,79]. As with MPNST detection, it may be that NF1 patients are less attuned to early skin changes of melanoma due to psychological desensitization from overall cutaneous burden of other NF1 findings. In the clinical observation of these authors, a portion of NF1 patients with less melanotic skin seek sun exposure to minimize the skin color irregularities associated with NF1. The incidence of sun-seeking behavior and its contribution to skin cancer in NF1 have not yet been formally investigated.

Landry et al. compared prevalence and disease-specific survival (DSS) rates for NF1 patients with skin cancer compared to skin cancer in the general population. In the NF1 cohort, 15 patients were diagnosed with melanoma (0.9%), which is over three times higher than the prevalence in the general population (0.24%). The age of diagnosis for NF1 patients was also lower, at 51.8 years compared to 64 years in the general population. Importantly, the 5-year DSS for NF1 patients was 66.7% compared to 92% in the general population. In addition, six (40%) of the NF1 patients diagnosed with melanoma had metastatic disease at diagnosis and five (33%) patients with localized disease developed metastasis post-surgery [78].

Guillot et al. analyzed 671 NF1 patients and found that 11 (1.6%) had melanoma, with a 10:1 female to male ratio. NF1 patients demonstrated a younger median age of occurrence (33 years) compared to the general population (46–53 years) [80]. Although there is some research conducted on skin cancers in NF1 patients compared to the general population, most focuses on the relative incidence and very few works specifically compare the outcome differences between NF1 patients and the general population. Future studies are needed to investigate these differences. For the general population, there have been significant advances in surgical approaches to skin cancers in recent years, reducing the invasiveness and morbidity associated while increasing the accuracy and efficacy of the resection. Advanced skin cancers can also be treated effectively utilizing immune checkpoint inhibitors such as ipilimumab and nivolumab in combination, which is associated with a response rate of over 50% and favorable survival rates [81].

While there are no current consensus guidelines or recommendations concerning skin cancer screening for NF1 patients, Hernandez-Martin et al. recommend routine annual follow-ups with a dermatologist given the increased risk for skin cancer, and the prognosis is majorly affected by the early diagnosis of malignant tumors [82,83]. Bergqvist et al. encouraged patients to visit an NF1 specialist every two to three years while having annual visits with dermatologists to follow up on cutaneous lesions to promote early detection and improve NF1 management [84]. Many NF clinics recommend annual appointments with the NF expert to look for tumors and other non-neoplastic sequelae of NF1. Pollack et al. analyzed 68,495 primary cases of melanoma diagnosed from 1992 to 2005 and found a significant difference between melanoma-specific survival (MSS) by stage at diagnosis, concluding with a recommendation for early detection through screening those at risk for melanoma, thus applying to NF1 patients [85]. A study analyzing 1561 biopsies in 1010 patients from three dermatology clinics affiliated with Loyola University Medical Center found that a dermatologist-performed skin screening first identified 797 skin cancers (51%), compared to 764 (49%) identified by the patient or referring provider. Additionally, melanoma identified through dermatology screening had a mean Breslow depth of 0.53 mm compared to the mean depth of 1.04 mm identified by the patient or a referring provider, providing justification for earlier detection via skin screenings in NF1 patients as a means to enhance survival and reduce the financial burden to both the patient and the healthcare system [86].

The improvement of skin cancer screening in NF1 patients is a multifactorial process. These patients would benefit from the implementation of frequent and longitudinal dermatologic care. Given the increased risk of melanoma and NMSC in patients with NF1, consensus screening guidelines are needed in order to promote early skin cancer detection in this high-risk population.

## 14. Cutaneous T Cell Lymphoma

Cutaneous T cell lymphoma (CTCL) is a helper T-cell neoplasm that manifests in the skin as a chronic, typically slowly progressive disease. The most common subtypes are mycosis fungoides and Sezary syndrome. Diagnosis is difficult given the various clinical presentations. Patients can present with erythematous and scaly patches, plaques, or tumors. Histopathology typically showed a thickened epidermis and a dense infiltration of lymphocytes in the dermis. The etiology is unknown; however, it is hypothesized that it may be, in part, due to chronic inflammation, from atopic dermatitis or contact dermatitis, for example. Patients can develop visceral organ involvement, and some individuals suffer with fulminant disease, leading to death. Treatment is not standardized and so requires an individualized approach. Early in the disease, patients are treated with skin-directed therapy, including topical corticosteroids, phototherapy, topical bexarotene, topical mechlorethamine, and localized radiotherapy. As the disease progresses, total skin electron beam therapy, systemic chemotherapy, interferon alpha, and systemic retinoids may be utilized.

Although no distinct relationship has been established, there have been multiple case reports of CTCL in patients with NF1 [87,88,89]. In these three case reports, two patients were diagnosed with mycosis fungoides [88,89] and one with a rare subtype of cutaneous follicular helper T-cell lymphoma [87]. While patients with NF1 are at higher risk of malignancies, there is no known predilection for CTCL specifically. One study analyzed next-generation sequencing data from 220 CTCL patients and found the NF1 gene to serve as one of several genetic drivers in these tumors, due it its amplification of the MAPK signaling pathway [90].

## 15. Wound Healing and Scarring

NF1 may confer a heavy burden of cutaneous manifestations requiring recurrent surgical interventions, wherein cosmetic outcomes are often a high priority. There is limited research currently regarding wound healing and the propensity for patients with NF1 to develop hypertrophic scars or keloids. Although current expert recommendations are for early resection of bothersome (symptomatic or unsightly) CNs, cosmetic outcomes are important factors when counseling patients on surgical options. In a recent small study investigating epidermal wound healing in NF1 patients, comparing five NF1 patients to six healthy controls, a suction blister was formed with a specialized device, unroofed, and dressed. Four days later, biopsies were performed for immunohistochemical analysis. There were no differences between NF1 patients and controls in the epidermal wound healing process, as scored with clinical evaluation and transepidermal water loss (TEWL). There were no differences between NF1 patients and controls in cell proliferation rates or Ras-MAPK activity in both keratinocytes and fibroblasts. This suggests that epidermal wound healing in NF1 patients is comparable to that of controls [91].

Moyawaki et al. conducted a retrospective study to determine whether surgical wounds in patients with NF1 were more likely to progress to hypertrophic scars or keloids. Cutaneous, subcutaneous, submucosal, and subfascial tumors were included and all were given a histopathologic diagnosis of CN. Wound healing was compared across 53 post-operative subjects with NF1 and 35 subjects with non-NF1-associated tumors. In the NF1 cohort, 0 of 53 patients developed a hypertrophic scar. In the solitary neurofibroma group, 2 of 35 patients (5.7%) developed a hypertrophic scar. No patients developed a keloid [92].

A large prospective study of 84 CNs resected from 12 NF1 patients included 11 who were white and one who was Black. After an average of five months of follow-up, patients had overall satisfying results without any skin infections, tumor regrowth, hypopigmentation, or keloid formation. One surgical site (1.2%) in the Black patient had a hypertrophic scar and 10 sites (12%) had post-inflammatory hyperpigmentation. Reported improvement in symptoms, daily activities, leisure, personal relationships, and treatment experience significantly increased post-surgical procedure (*p* = 0.00062) [29]. These data suggest that patients with NF1 may not be at higher risk for keloid or hypertrophic scar formation compared to the general population. However, future research is needed to subdivide risk by ethnicity or skin type and to identify predictive factors for clinical counseling and patient management. Of note, in the general population, keloids are most commonly seen in individuals of African, Asian, Hispanic, and Mediterranean descent. The incidence of keloid development in these populations is approximately 4.5–16% [93]. Individuals with more melanotic skin tend to develop keloids 15 times more frequently when compared to their lighter-skinned counterparts [94]. While the propensity of NF1 patients to develop keloids is not well studied, racial predilection should be taken into account for optimal patient guidance.

## 16. Conclusions

NF1 confers risks of a broad range of cutaneous manifestations and requires an interdisciplinary approach to best manage patients. Although the classical findings of café-au-lait macules, flexural freckling, and CNs are criteria for a clinical diagnosis of NF1, additional conditions including glomus tumors, JXG, and skin cancer are important for clinical counseling and monitoring. Current options for CN removal include surgical removal, laser and light procedures, and electrodessication. Given that CNs are often the most burdensome symptom for NF1 patients, future research is needed to explore safe and effective therapies.

## Figures and Tables

**Figure 1 cancers-15-02770-f001:**
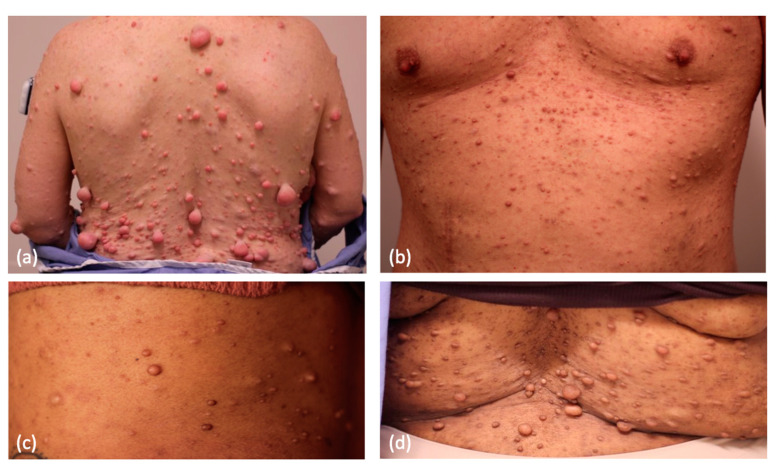
Cutaneous neurofibromas in NF1 patients with varying Fitzpatrick skin phototypes: (**a**) Fitzpatrick II, (**b**) Fitzpatrick III, (**c**) Fitzpatrick IV, and (**d**) Fitzpatrick V.

**Figure 2 cancers-15-02770-f002:**
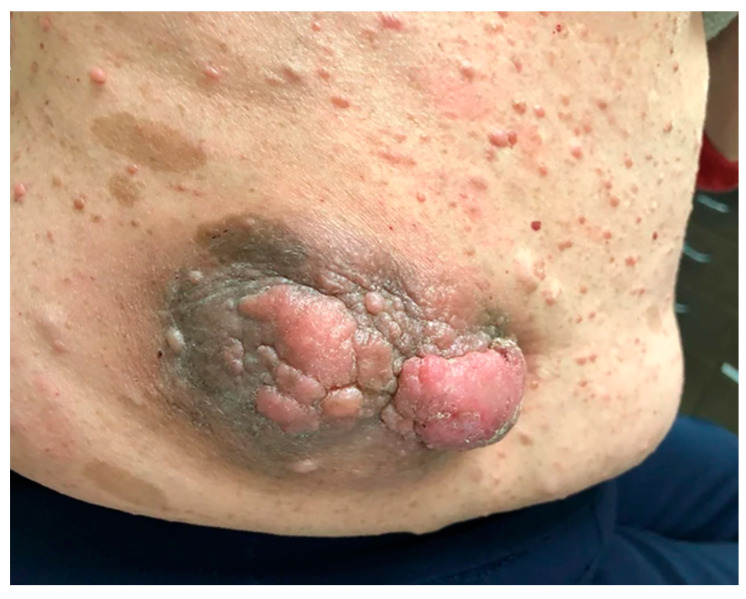
An MPNST with dermal involvement on the right flank of an NF1 patient [50].

**Figure 3 cancers-15-02770-f003:**
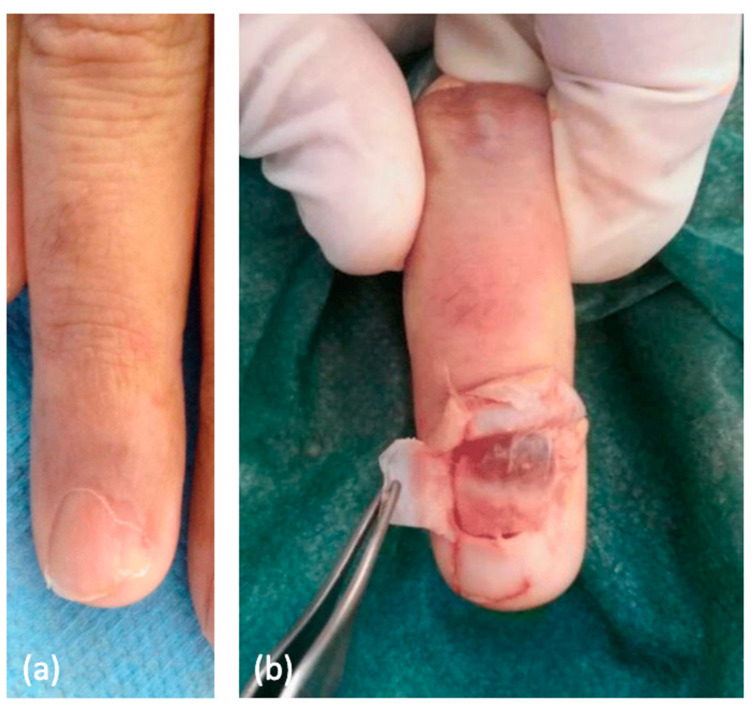
(**a**) A glomus tumor presenting nail dystrophy. (**b**) Surgery revealed a red-blue subungual glomus tumor [60].

**Table 1 cancers-15-02770-t001:** Systemic and topical treatments for cutaneous neurofibromas.

	Drug	Mechanism of Action	Study Population	Results	Side Effects	Status
** *Systemic Treatments* **						
	Imatinib mesylate 400 mg daily [18]	C-kit receptor tyrosine kinase inhibitor	1 adult patient	Small reduction in visual burden	Fluid retention, gastrointestinal symptoms, hepatotoxicity, fatigue, rash	N/A ^a^
	Everolimus 5–15 ng/mL serum concentration [17]	mTOR ^b^ inhibitor	17 adult patients	Significant reduction in absolute value of paired lesion height and significant reduction in lesion surface volume in 3/17 patients	Gastrointestinal symptoms, upper respiratory infections, skin irritation	Complete (NCT02332902)
	Selumetinib ^c^ [23]	MEK ^d^ 1/2 inhibitor	Adult patients	N/A	N/A	Active, not recruiting (NCT02839720)
** *Topical Treatments* **						
	Imiquimod 5% [24,25]	Toll-like receptor 7 agonist	11 adult patients	No significant reduction in tumor volume	Skin irritation and erythema	Complete (NCT00865644)
	Diphen-cyprone 0.04% [28]	Hapten that induces a delayed-type hypersensitivity reacion	Adult patients	N/A	Skin irritation and erythema	Active and recruiting (NCT05438290)
	NFX-179 0.05%, 0.15%, 0.50% [26]	MEK inhibitor	47 adult patients (35 in the treatment arms)	−1.6, −11.9, and −16.7, percent changes in CN volume for 0.05%, 0.15%, and 0.50% concentrations, respectively	Pruritis, stinging, erythema	Complete (NCT04435665)
	NFX-179 0.5%, 1.5% [27]	MEK inhibitor	Adult patients	N/A	N/A	Active, not recruiting (NCT05005845)

^a^ This study was a case report. ^b^ Mammalian target of rapamycin. ^c^ Dose not specified. ^d^ Mitogen-activated protein kinase.

**Table 2 cancers-15-02770-t002:** Procedural treatments for cutaneous neurofibromas.

	Mechanism	Amount of CNs ^a^ That Can Be Removed in One Session	Anesthesia	Benefits	Risks and Limitations
**Electrodessication**	Radiofrequency ablation	100–1000 s	Local or general anesthesia	Time-sparing, low cost, does not require specialized providers, low rates of post-op bleeding, deep penetration, ideal for small tumors	Thermal necrosis of surrounding normal skin, increased scarring [30]
**Traditional Surgical Removal**	Excision	10–100 s	Often requires general anesthesia	Good cosmetic outcomes, linear scarring, ideal for large tumors > 4 cm	Expensive, time-intensive, requires trained surgical specialists, higher risk of adverse events, requires sterile technique, higher risk of bleeding, difficult to remove small CNs, requires suture removal [25,29]
**Modified Biopsy Removal**	Excision	100 s	Local anesthesia	Good cosmetic outcomes, low cost, does not require specialized providers, outpatient setting, nonsterile technique, prevents tumor regrowth due to dermal removal	Requires suture removal [29,30]
**CO_2_ ^b^ Laser**	Thermal ablation	100 s	Local or general anesthesia	Time-sparing, immediate hemostasis, healing by secondary intention, improved symptoms of CNs, ideal for small tumors	Expensive, requires trained specialists, scarring (often hypopigmented, atrophic, or hypertrophic), less penetration (better-suited for superficial CNs) [33,35,41]
**Er:YAG ^c^ Laser**	Thermal ablation	100 s	Local or general anesthesia	Time-sparing, rapid re-epithelialization, decreased duration of post-op erythema, shorter recovery, less edema, decreased thermal necrosis, greater precision, good cosmetic outcomes, ideal for small tumors	Expensive, requires trained specialists, less hemostasis than CO_2_ laser [41]
**Nd:YAG ^d^ Laser**	Thermal ablation	100 s	Local or general anesthesia	Time-sparing, decreased post-op hypopigmentation, preserves epidermis, high penetration, ideal for small tumors	Expensive, requires trained specialists [42]

^a^ Cutaneous neurofibromas. ^b^ Carbon dioxide. ^c^ Erbium-doped yttrium aluminum garnet. ^d^ Neodymium-doped yttrium aluminum garnet.

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
