# Peer review of "Dermatologic Manifestations of Neurofibromatosis Type 1 and Emerging Treatments"

_cancers, 2023, doi:10.3390/cancers15102770_

Round 1

Reviewer 1 Report

The authors present a review of NF1 dermatologic manifestation and  emerging treatment options. The manuscripts includes cutaneous neuro- 22 fibromas, plexiform neurofibromas, diffuse neurofibromas, distinct nodular lesions, malignant pe- 23 ripheral nerve sheath tumors, glomus tumors, juvenile xanthogranulomas, skin cancer, and cutane- 24 ous T-cell lymphoma. The authors discuss and present topical, systemic and laser, light based and  surgical treatment options as well.

The manuscript is well written and the scientific aspect is well discussed. However, I would suggest some additional aspects to improve the content for the readers:

- please add a table of different studies and results

- discuss the results of previous publications regarding outcome and prognosis after treatment

Reviewer 2 Report

The authors have highlighted the role of Neurofibromatosis Type 1 (NF1) in various skin-related diseases including  cutaneous neurofibromas, plexiform neurofibromas, diffuse neurofibromas, distinct nodular lesions, malignant peripheral nerve sheath tumors, glomus tumors, juvenile xanthogranulomas, skin cancer, and cutaneous T-cell lymphoma. The article covered all the mentioned aspects very elegantly with illustrations. This is an excellent article in the field and will be beneficial to the audience of this area. 

Reviewer 3 Report

Dear, All,

The manuscript reviews neurofibromatosis type 1 concerning limited therapeutic options and emerging therapeutics. The review covers the full spectrum of the aspects of neurofibromatosis type 1. The manuscript has minor revisions. 

Minor revision.

(1) Lines 60-61 Sentence: “Genetic testing is not required for the diagnosis of NF1 but may contribute to a diagnosis.” The sentence needs clarification regarding the subject.
